# Dog Ecology and Rabies Knowledge of Owners and Non-Owners in Sanur, A Sub-District of the Indonesian Island Province of Bali

**DOI:** 10.3390/ani8070112

**Published:** 2018-07-05

**Authors:** Elly Hiby, Kadek Karang Agustina, Kate Nattras Atema, Gusti Ngurah Bagus, Janice Girardi, Mike Harfoot, Yacinta Haryono, Lex Hiby, Hendra Irawan, Levin Kalalo, Sang Gede Purnama, I. Made Subrata, Ida Bagus Ngurah Swacita, Ni Wayan Arya Utami, Pande Putu Januraga, Dewa Nyoman Wirawan

**Affiliations:** 1IFAW, Yarmouth Port, MA 02675, USA; katema@ifaw.org; 2Department of Veterinary Public Health, Faculty of Veterinary Medicine, Udayana University, Jln. PB Sudirman, Denpasar, Bali 80232, Indonesia; k.agustina@unud.ac.id (K.K.A.); ibnswacita@unud.ac.id (I.B.N.S.); 3BAWA, Jl. Anak Agung Gede Rai no. 550, Br. Kertha Wangsa, Lodtunduh, Ubud, Gianyar, Bali 80232, Indonesia; bagus@bawabali.com (G.N.B.); janice@bawabali.com (J.G.); cinta@bawabali.com (Y.H.); hendra@bawabali.com (H.I.); levin@bawabali.com (L.K.); 4UNEP-WCMC, 219 Huntingdon Rd, Cambridge, CB3 0DL, UK; mike.harfoot@unep-wcmc.org; 5Conservation Research Ltd., Cambridge, CB22 5AL, UK, lexhiby@gmail.com; 6Public Health Department, Faculty of Medicine, Udayana University, Jln. PB Sudirman, Denpasar, Bali 80232, Indonesia; sangpurnama@unud.ac.id (S.G.P.); madesubrata11@gmail.com (I.M.S.); arya.utami@unud.ac.id (N.W.A.U.); januraga@unud.ac.id (P.P.J.); wirawandewa48@gmail.com (D.N.W.)

**Keywords:** dog, canine, stray dog, demographics, rabies, vaccination, owner behaviour, owner knowledge, Bali.

## Abstract

**Simple Summary:**

That rabies can be managed humanely using vaccination is well accepted. However, making sure enough dogs in the population are vaccinated and therefore able to fight off the virus can be challenging. Getting owners to be more responsible for their dogs should help keep the proportion of vaccinated dogs high. This study looked at dogs and people living in three villages in Sanur, Bali; the total population of dogs was 6009, of which only 200 dogs appeared to have no owner. There were some differences between these 6009 dogs in terms of their welfare, the likelihood of them being unvaccinated, the method owners used to confine them, whether they were sterilised, and where owners got them from. Most people were well informed about rabies and had positive attitudes towards dogs and vaccination. This information could then be used to plan a project to improve responsible dog ownership.

**Abstract:**

This study gained an understanding of dog demographics, owner behaviour, and knowledge, attitudes and practices relating to rabies in three villages in Sanur, Bali, providing insights for an intervention to improve responsible dog ownership. A combination of a census of all dogs, street surveys of roaming dogs, and a Knowledge, Attitude and Practices (KAP) survey was used to study both dogs and people. A total of 6009 dogs were recorded, of which only 3.3% appeared to be unowned; unowned dogs had poorer welfare and were more likely to be wary of attempted approaches by people. The source of dogs, method of confinement used by owners, and whether dogs were sterilised differed between three breed types; purebred dogs, mixed breed, and Bali dogs (native breed). Three variables were found to have significant effects on the chances of not being vaccinated: age, dog type, and confinement. A mean of 3.81 roaming dogs per km of street surveyed was recorded along 28 sub-village routes. Responses to attitude statements showed that most people had a positive attitude towards dogs and vaccination and did not agree with culling. Knowledge of appropriate bite treatment and symptoms of rabies in dogs was good.

## 1. Introduction

Rabies is a fatal viral disease, causing an estimated 59,000 human deaths worldwide each year [1]. It is predominately transmitted to humans from bites of infected domestic dogs [2]. There is significant evidence [3] and widespread international support [4] for the approach of eliminating rabies through annual mass vaccination of at least 70% of the domestic dog population. In comparison, evidence suggests that culling is ineffective [5,6,7]; this approach has failed to eliminate the virus in practice, e.g., in Flores, Indonesia [8] and it is a widely unpopular control method with local people [9].

Rabies has slowly spread through Indonesia and is currently endemic in 24 of the 33 Indonesian provinces; the Indonesian island province of Bali was historically rabies-free until an incursion in 2008 led to virus spread across the island. In the subsequent years, many thousands of people received costly post-exposure prophylaxis following dog bites and over one hundred people died [10]. The animal cost has also been high, with many hundreds of reported deaths from rabies and many thousands culled in misguided attempts to stop the spread [11].

Following concerted efforts by local, national and international agencies, several years of mass dog rabies vaccination have been implemented in Bali with resulting decreases in the number of both dog and human cases of the disease [12]. However, population turnover resulting from births of susceptible puppies and deaths of vaccinated dogs, plus the import of unvaccinated dogs, can challenge rabies control by reducing vaccination coverage below critical levels [13]. Effective control can also be diminished by pockets of unvaccinated dog populations, which can provide reservoirs for the virus [7].

Although mass vaccination of dogs is now the accepted policy for rabies control in Indonesia, sporadic culling does still occur in Bali and arguably exacerbates the challenges of population turnover and replacement of culled dogs with unvaccinated individuals. In addition, irresponsible dog ownership practices, such as failure to access vaccination services and the abandonment of unwanted litters of puppies and adult dogs, reduces vaccination coverage further.

It is proposed that improved maintenance of vaccination coverage can be achieved through increased responsible ownership of dogs by increasing uptake of vaccination services, reducing unwanted litters, and building closer owner–dog relationships to reduce abandonment and improve dog care. The aim of this study was to gain an understanding of the current dog population, local ownership practices, and knowledge of rabies control and attitudes to allow for targeted design of interventions to improve responsible ownership. The results of a dog population census, the Knowledge, Attitude and Practices survey, and a street survey are presented here, covering three *desas* (villages) within Bali, with subsequent implications for future management.

## 2. Materials and Methods

The boundaries of three neighbouring *desas* were identified as the region of interest: Sanur Kaja, Sanur Kauh, and Kelurahan Sanur, with a total human population of 24,373 (local government statistics reported as a pers. comm. from government representative). These *desas* reported a total of eight human rabies deaths in 2009 and 2010 (local government statistics reported as a pers. comm. from a government representative). There have been no reported human cases since 2010, and the last reported and confirmed dog rabies case was in April 2012. However, as rabies continues to be reported in other areas of Bali, the risk of rabies entering these areas via an infected dog is still present. The Sanur area of Bali is densely populated with people, is perceived to have a high roaming dog population, and is popular with tourists who sometimes feed roaming dogs; these factors contribute to the local government and communities being particularly motivated to engage in an intervention to improve rabies control through responsible dog ownership. The dog population and people living within the region of interest were studied using three methods: a dog demography questionnaire, a Knowledge, Attitude, and Practices questionnaire, and a street survey.

### 2.1. Dog Demography Questionnaire

Teams of veterinarians and public health professionals (termed ‘T1’s because they became the first tier of trainers in the subsequent project) conducted a complete census of all dogs living within the three *desas* from May 2016 to May 2017; this interview with dog owners and carers of unowned dogs was also used to engage people on the subject of dog care and welfare. Interviews were conducted throughout the year at all times of the day, sometimes requiring more than one visit to a household to meet the owner at a convenient time. The census included dogs living within households at that time (it was not retrospective) and those residing in public places with no identifiable owner, but commonly benefiting from care provided by sympathetic local people. Dog-owning households were identified through a combination of exhaustive searching of all streets and snowballing from one dog-owning household to neighbouring households with dogs. People were also asked to identify locations where unowned dogs were known to live; when apparently unowned dogs were found, people living and working locally were asked to confirm whether these dogs were indeed unowned or owned dogs roaming unsupervised. Photographs of each dog were taken and used to ensure dogs were not double counted; this was a potential risk with loosely owned dogs that may be claimed by more than one household or perceived as unowned when seen roaming. Bali dogs have a relatively varied phenotype hence this catalogue of photos appeared to be an effective mitigating measure. T1s were also allocated *banjars* (smaller administrative areas within *desas*) to complete by themselves or in pairs, to reduce the risk of double counting, as they became familiar with all the dogs in the *banjar* as the interviews progressed.

A dog demography questionnaire was conducted for each dog in every dog-owning household. A similar questionnaire was completed for all unowned dogs that could be found, with obvious limitations on the questions that could be answered when there was no owner. The questionnaire was formed of closed questions exploring gender, age, type, source, sterilisation, vaccination, breeding condition in the case of females, and level of confinement; owners were asked to state the confinement method used for the majority of the dog’s average day, including house and/or yard (defined as an outdoor space with some kind of physical boundary, usually a wall), kennel or cage, and tether or chain. Whether this method of confinement was effective, e.g., whether the yard boundary was truly dog-proof, was not investigated during the interview. It is possible that some of the dogs reported by their owners to be confined could potentially escape. Observation of the dog was used to assess visible welfare status in terms of body condition (using the body condition score scale available at http://www.icam-coalition.org/IndicatorsProject.html), presence or absence of a visible skin condition, and presence or absence of visible wounds. The questionnaire provided in the annex to the International Companion Animal Management (ICAM) Coalition’s ‘Are we making a difference?’ guide was used as a starting point [14]; however, to shorten the time required to complete the interview, only those questions that appeared most relevant to responsible ownership and vaccination of Balinese dogs were selected by the authors for this household questionnaire. It also included a novel test for the dog’s response to being called over and petted, first by the owner (if an owned dog) and then the interviewer; the dog’s responses to both attempts were scored as ignores, wary, aggressive, positive/friendly or refusal (where either the owner or the interviewer refused to try). The questionnaire data was recorded using either of two smart phone apps: Epi Info™ (a database and statistics program for public health professionals, developed by the Centers for Disease Control CDC, Atlanta, GA, USA, https://www.cdc.gov/epiinfo) or Device Magic (a mobile phone application for data collection, https://www.devicemagic.com), in combination with Wise Monkey (a web-based database and data management system that works in tandem with mobile phone applications, developed by the Wise Monkey Foundation, http://www.wisemonkeyfoundation.org/). The list of questions and possible responses is available in the Appendix A.

### 2.2. Knowledge, Attitude and Practices (KAP) Questionnaire

In addition to the dog demography questionnaire, a second questionnaire exploring knowledge and practice relating to rabies and attitudes towards dogs and methods of rabies control was delivered to all dog owners and a random sample of at least 10 non-dog owning households per *banjar*. This included Likert scales following attitude statements, selection of a preferred outcome following a scenario and three open questions exploring knowledge about what to do if bitten by a dog, how often a dog should be vaccinated and what the symptoms of rabies would look like in a dog. The list of questions and possible responses is available in the Appendix A.

### 2.3. Street Survey

Street surveys using direct observation of roaming dogs on public roads were conducted along routes within all *banjars*. These routes were designed to cover as many of the streets as possible within a *banjar* (at least half of the streets) but following an efficient route that minimised retracing previously surveyed streets. Surveyors worked in pairs, travelling on motorbikes with the driver navigating along the route and the passenger observing and recording the presence of roaming dogs using a smart phone app called OSMTracker (an Open Street Map tracker application that records the GPS location of points of interest, in this case, the presence of roaming dogs). These surveys were not intended to produce an estimate of the total number of roaming dogs, instead the length of route and number of dogs observed was recorded by the app, resulting in a measure of density expressed as the number of roaming dogs observed per km of street surveyed; the route was carefully recorded to allow the survey to be repeated using the same route and protocol in future, with the aim of monitoring of roaming dog density over time [15]. Each route was surveyed two or three times on consecutive days to establish day-to-day variation in roaming dog density. Surveys were conducted in the early morning between 5 and 8 a.m.; at this time traffic is light and so roaming dogs tend to be more easily visible and routes can be travelled more efficiently. All *banjars* within a *desa* were surveyed within a 2–3 week period; this *desa*-focused survey event was triggered by nearing completion of the dog demography questionnaire for all *banjars* in that *desa*. As the *desas* were studied consecutively; this meant the street surveys were conducted in July/August 2016, October/November 2016, and February/March 2017.

### 2.4. Data Storage and Analysis

Questionnaire data was stored on a secure password protected website and analysed using the Epi Info Visual Dashboard (a module of the Epi Info™ 7 program). Street survey data was stored on an offline Microsoft Access database and analysed using Microsoft Excel.

Dog demography data was analysed as percentages of the dog population falling into different categories of variables. The effect of dog characteristics such as gender and breed were tested for effect on management of the dogs, such as whether the dog was sterilised or confined using Chi-squared tests. The significance and effect size of selected dog characteristics on the odds of not being vaccinated was explored using logistic regression; this was conducted in R [16] using R package ‘multcomp’ to fit a general linear model (GLM) with binomial error distribution and a logit link function.

The KAP questionnaire data was analysed as percentages of interviews responding in particular ways. The effect of being a dog owner versus a non-owner on responses to questions was tested using Chi-squared tests.

The street survey data was analysed as the mean number of roaming dogs observed per km of street surveyed, the variance presented is the average variance resulting from the within route two or three replicates on consecutive days.

### 2.5. Ethics

Respondents were first asked whether they agreed to take part in the study. An introductory statement explained the purpose of the study and that information about the dogs would be shared within the study team. If data was shared outside the team it would be anonymised. Respondents could refuse to take part at any point and their data would then be removed from the dataset.

The study initially began as programmatic work and was not presented to an ethics committee at the outset. However, once the research potential of the data was realised, the protocol was presented and approved by the Ethics Committee of Faculty of Veterinary Medicine Udayana University, No. 356/KE-PH-Lit-3/II/2018; no amendments to the protocol were suggested by the Ethics committee. All data collection, before and after approval was given, was conducted in accordance with the Declaration of Helsinki.

Data was stored on a secure server. A limited number of team members had access to the raw data and this access was password-protected.

## 3. Results

*Desas* were homogenous in their dog population characteristics and participant responses and hence data was combined across the three *desa* populations.

### 3.1. Dog Demography

The census included a total of 6009 dogs; 96.7% of these dogs had an identifiable owner (5809) as only 200 unowned dogs could be found, representing 3.3% of the total dog population. There was an estimated ratio of one dog to every four people. The 5809 owned dogs were owned by 3093 owners, resulting in an average of 1.88 dogs per dog owner. As some households in Sanur are formed of extended families living in several dwellings within one household compound, it is possible for a household to have more than one dog owner, hence this does not equate exactly to an average number of dogs per household.

Gender ratio was close to equal, with 46.2% females, or 1.17 males to each female. However, there was a difference in gender ratios between owned and unowned dogs, with a skew towards females in unowned dogs (55.3% females) and a skew towards males in owned dogs (54.1% males). This was a significant trend at the 10% level (*X*^2^ = 6.523, *p* = 0.0106). At the time of the joining the census population, 4.9% of the females were lactating, 4.6% were reported to be pregnant, and 6.1% were reported to have had puppies in the last 12 months.

Dogs were categorised into three breed types: 22.5% Bali dogs (native breed), 45.2% mixed breed, or 32.3% purebred dogs. These different dog breed types had significantly different sources (see Figure 1); adoption from the street was the most common source for a Bali dog, being born into the household as the pup of another household dog was the most common source for a mixed breed dog, and purchase from outside the *desa* was the most common source for purebred dogs (*X*^2^ = 1080.32, *df* = 18, *p* < 0.001).

Figure 2 shows the age histogram of all the dogs. The largest age group was that aged under one year, with 22.8% of dogs falling into this youngest category. Old dogs were defined as those of seven years or older—10.6% of the dogs fell into this category. The maximum reported age was 20 years.

### 3.2. Confinement

Most dogs were confined to either a house or a yard (49.9%). The next most common methods of confinement were no confinement at all, with 19.4% of dogs just being allowed to roam freely, and an equal percentage was confined to a kennel or cage. The least most common form of confinement was being tethered (11.2% of all dogs).

There is a significant effect of breed type on method of confinement with being allowed to roam freely as the most common way of keeping a Bali dog, whilst confinement to a kennel or cage being the most common method for confining purebred dogs (*X*^2^ = 700.608, *df* = 6, *p* < 0.001; see Figure 3).

### 3.3. Sterilisation.

Owners and carers of unowned dogs reported that 20.9% of dogs had been sterilised; there was no effect of gender on the chance of being sterilised (*X*^2^ = 6.036, *p* = 0.014). However, there was a significant effect of breed type on the chances of being sterilised, with 35.9% of Bali dogs having been sterilised as compared to just 19.0% mixed breed dogs and 14.7% purebred dogs (*X*^2^ = 244.009, *df* = 2, *p* < 0.001; see Figure 4).

### 3.4. Vaccination

Owners and carers of unowned dogs reported that the majority of dogs were vaccinated against rabies (57.7%). For 53 of these vaccinated dogs, their date of last vaccination was greater than one year and hence their vaccination status was categorised as ‘lapsed’, resulting in 56.0% of dogs with a current vaccination status.

Four variables were selected to test for their ability to predict whether a dog was likely to be unvaccinated; gender, age, breed, and confinement method. These variables were selected because they were all immediately apparent by simply observing the dog, as opposed to variables such as source of the dog or prior breeding history of females which would require owner interview. This would potentially allow future vaccination campaign staff to target dogs likely to be unvaccinated based on visible dog characteristics. Prior to running the logistic regression, age was split into four categories: puppy (<4 months), juvenile (4–12 months), adult (1 < 7 years), and old adult (7+ years). Dogs with incomplete data for any of the four variables or vaccination status were removed from the analysis, leaving *n* = 5461.

Logistic regression was used to test both main effects and pairwise interactions to establish the odds of not being vaccinated for each variable and pair of variables. Only one pairwise interaction was found to be significant; Bali dogs allowed to roam freely without confinement had an odds ratio of 1.597 of being unvaccinated. The pairwise interaction model had an Akaike Information Criterion (AIC; an estimator of the relative fit of the statistical model to the data) value of 6761.3, whilst the main effects model had a lower AIC of 6728.5. Thus, as well as there being only one significant pairwise interaction, the AIC of the pairwise model suggested it was not as good at explaining the variation in the data as the main effects model. Hence, the following results are from the main effects model (Table 1).

Age was found to have the most significant effect on vaccination; as dogs got older they became more likely to be vaccinated. Puppies had an odds ratio of 10.951; in other words, puppies were over 10 times more likely to be unvaccinated than old adults of seven years or older. Juveniles were also more likely to be unvaccinated, with an odds ratio of 1.908, and as such were nearly twice as likely to be unvaccinated than old adults. Adults had similar odds of being vaccinated to old adults. Dog type was found to be significant; the odds ratio for mixed breed dogs being not vaccinated was 1.276 and for Bali dogs the odds ratio was 1.348 as compared to purebred dogs. Confinement method was found to be significant; the odds ratio of dogs allowed to roam freely not being vaccinated was 1.286 compared to those confined to a house or yard. Gender was found to be a non-significant predictor of vaccination.

### 3.5. Welfare Status

Most dogs were observed to have a body condition score of 3 (ideal), however 6.2% of owned dogs and 23.2% of unowned dogs had a body condition score of 1 (emaciated) or 2 (thin); there was a statically significant relationship between ownership status and body condition score (*X*^2^ = 91.758, *df* = 4, *p* < 0.001) (Figure 5).

Most dogs had no visible skin condition, however 11.1% of owned dogs and 18.8% of unowned dogs did have a visible skin problem; there was a statically significant relationship between ownership status and skin condition (*X*^2^ = 11.195, *p* < 0.001) (Figure 6).

Most dogs had no visible injuries, however 4.4% of owned dogs and 10% of unowned dogs did have a visible injury; this was a statically significant relationship between ownership status and visible injury (*X*^2^ = 13.578, *p* < 0.001) (Figure 7).

### 3.6. Response to Owners and T1s

For owned dogs, both the owner and T1 called the dog over and attempted to pet them; for unowned dogs this response test was only done by the T1. In the vast majority of cases the response was positive (77.3%), the second most common category of response was wary (12.3%), and there were few aggressive responses (5.8%) (Figure 8). When looking at only the owned dogs, they responded more positively to their owners and showed more wary and aggressive responses to the T1s, who would have been strangers to the dog (*X*^2^ = 1951.77, *df* = 4, *p* < 0.001).

Comparing owned and unowned dogs in their response to T1s, owned dogs showed more positive responses to these strangers, whilst unowned dogs were more likely to be wary or ignore the T1′s attempts to call them over (*X*^2^ = 61.705, *df* = 4, *p* < 0.001) (Figure 9).

### 3.7. Knowledge, Attitude and Practices Questionnaire

The KAP survey was delivered to 3171 people; 2594 dog owners, 516 non-owners and 61 where dog ownership status was not recorded. The following shows the levels of agreement with each of the attitude statements for both dog owners and non-owners.

In the diverging stacked bar charts that follow (Figure 10 and Figure 11), blue bars indicate agreement with positive statements or disagreement with negative statements. Red bars indicate disagreement with positive statements and agreement with negative statements. Hence, the more blue in the chart, the more positive the attitudes towards dogs and vaccination rather than culling. In the charts the attitude statements have been shortened to key phrases; the following table provides the full attitude statement for each key phrase (Table 2).

Responses to attitude statements showed that most people had a positive attitude towards dogs and vaccination and did not agree with culling. For all attitude statements, dog owners showed significantly more positive attitudes towards dogs and vaccination than non-dog owners.

Respondents were asked “imagine that a dog belonging to a family in your *banjar* becomes sick. It is thin, it has lost fur and its skin looks bad. What do you think should happen?” Most people (48.3%) responded “the family is responsible for getting advice on how to care for the dog, to see if it can be made better”. There was a significant difference between the responses of owners compared to non-owners (*X*^2^ = 62.853, *df* = 5, *p* < 0.001), as shown in Figure 12.

Respondents were asked how they would respond “if they were bitten by a dog they did not know and therefore did not know the vaccination status?” Most people described correct responses; the most common was to attend the bite clinic or hospital, as mentioned by 73.9% of people. Of the incorrect responses, using traditional medicine to treat the bite would was the most common, as reported by 7.5% of people (Figure 13).

Respondents were asked to list symptoms of rabies in dogs. Most reported correct symptoms, with 40.3% mentioning excessive salivating or drooling, although only 9.2% mentioned the unique rabies symptom of aggression or unprovoked biting. Of the incorrect responses, 2.9% of people mentioned provoked aggression in defence of self or property, whilst 1.9% and 1.8% mentioned skinny and bad skin, respectively (Figure 14).

### 3.8. Street Survey

Street surveys were conducted in 28 *banjars* along routes of an average 6.4km in length (Figure 15). A mean of 3.81 roaming dogs was observed per km of street surveyed, with an average within route variance of 1.56 roaming dogs/km.

## 4. Discussion

The combination of the dog demography questionnaire, KAP survey, and street survey was able to expose a detailed picture of the dogs and owners living in this community in Bali. Taking the approach of a complete census with individual dogs as the sample unit (as opposed to households) in which all dogs living within the community were included in the dog demography questionnaire was time-consuming. It took one year for all three *desas* to be studied by a team of approximately 10 people. However, this intensive approach yielded two benefits: (1) The opportunity to meet all dog owners and carers in the community and identify those that required follow-up support such as vaccination or sterilisation services; and (2) the census of all dogs allowed for estimating the total owned (5809; 96.7%) and unowned (200; 3.3%) dog population. This differs from an earlier study in Bali [12] and another on Lombok (a neighbouring Indonesian island)[17], where questionnaires on dog demography were carried out using households as the sampling unit and did not include unowned dogs. Although observations of free roaming dogs were also conducted in these studies, the composition of owned roaming and unowned dogs in these free-roaming populations was not calculable [12,17].

Although every effort was made to identify every dog in all three *desas*, it is feasible that some dogs were missed from the census and these would potentially be biased towards unowned dogs as there was no referral household to report their existence. However the proportion of unowned dogs found is higher than a previous census study conducted in two other *desas* in Bali which identified between 1.1 and 1.5% unowned dogs [18], giving confidence that few if any unowned dogs were missed.

Because the dog demography questionnaire provided an estimate of the total owned and unowned dog population, the method employed for the street survey focused on establishing baselines for roaming dog density per km of street surveyed [15]. This method requires fewer resources to complete as compared to mark–resight methods that aim to produce an estimate of the abundance of roaming dogs [19,20]. Hence, it can be repeated more frequently to monitor change in the roaming dog density over time and in response to any intervention. By monitoring roaming dog density, we do not wish to imply that the presence of roaming dogs precludes effective rabies control. On the contrary, it can be hypothesised that the presence of vaccinated roaming dogs may act as a barrier to the movement of symptomatic rabid dogs through a *banjar*, as these rabid dogs may elicit defensive or territorial aggression in resident dogs. Although we are not aware of objective testing of such a hypothesis, this could provide an example mechanism for the concept of “the vaccinated dog (as) the soldier in the fight against rabies” [21] (p. 4). However, roaming behaviour of owned dogs can be used as an indicator of responsible ownership, as greater food provisioning by owners and reduction in oestrus females through sterilisation should reduce motivations for roaming. The average density of dogs observed in this Balinese community (3.81 dogs per km of street surveyed) was similar to that seen in communities in Latin America and Europe, but lower than the density observed in Kathmandu, Nepal [15]. The variation in the density of dogs observed on the same route, but on consecutive days, was relatively low suggesting, that this will provide a sensitive indicator to any changes in dog density over time.

The results of the dog demography questionnaire can be compared to similar studies of dog populations in Bali [12,18] and in Lombok [17]. That females represent a minority of owned dogs was consistent across all four studies. The composition of the dog population in terms of breed type was similar in our study to Lombok, with approximately one-third of dogs being of an identifiable breed, excluding the native Bali dog breed, which is the most common breed type in our study. Breed type was not reported for the other two Bali studies [12,18]. However, confinement of dogs appears to differ in our study, with a lower proportion of dogs described as never confined and always roaming (19.4%), as compared to 66% in the Arief et al. Bali study and ranging from 70.7% in the urban areas to 100% allowed to roam in the rural areas of the Lombok study. However, this may have been due to questionnaire design; we explored the method of confinement used for the majority of a dog’s day and did not ask owners to clarify if this method of confinement was permanent or temporary. Hence, many of the dogs reported as being confined to a house, yard, kennel, cage, or tether may also have roamed for part of the day.

The odds of not being vaccinated was explored for four variables considered to be immediately apparent on observation of a dog; gender, age, breed, and confinement method. As we aimed to identify visible predictors of being unvaccinated, allowing the future intervention to target dogs on sight. Age was established as the most significant factor for predicting whether a dog was unvaccinated; the older the dog, the more likely it was to be vaccinated, with puppies over 10 times more likely to be unvaccinated than old adults (7+ years of age). The same increased risk of not being vaccinated when young was found by Arief et al. [12]. Annual vaccination campaigns have been used in Bali to control rabies; dogs born since the last campaign will remain unvaccinated until the next round of vaccinations. This finding highlighted the potential benefit of using the subsequent intervention to increase access to vaccination between campaigns. It also suggests a need to explore owner perceptions of vaccinating puppies, in case there is misconception about rabies vaccine only being suitable above a minimum age. Also in agreement with Arief et al. was the predictive value of roaming behaviour; the odds ratio of dogs allowed to roam freely not being vaccinated was 1.286 compared to those confined to a house or yard. Although roaming behaviour itself may not prevent vaccination, it may indicate a lower level of responsible ownership, suggesting a focus on increasing a sense of responsibility over dogs and understanding of appropriate owner behaviours would be valid part of the future intervention. Areif et al. found female dogs were more likely to not be vaccinated, however gender was not a significant predictor in our population; this may be related to the greater gender skew towards males in the Arief et al. study as compared to our population. An additional significant predictor of vaccination in our population was breed type, with native Bali dogs and mixed breed dogs more likely to be unvaccinated than purebred dogs.

The dogs’ responses to being called over and petted by both owner (where available) and the T1 was difficult to standardise in the field. The available member of the household was interviewed for the dog demography questionnaire, and the dog’s response may have been different with respect to different members of the household. However, this provided an indication of how handle-able the dogs were, and hence the likelihood of expert catchers and handlers being needed in any subsequent vaccination campaigns. Nearly 95% of owned dogs showed positive responses to their owners and were more positive towards owners than T1s, suggesting that future efforts to vaccinate owned dogs should emphasise the involvement of owners and would rarely need expert dog handlers. Although many unowned dogs showed positive responses (41%), the majority did not, suggesting that it was when vaccinating these dogs that expert handlers and catchers would be needed.

The KAP survey focused on attitudes and knowledge of dogs and rabies. A similar KAP survey used in Ethiopia (Tschopp et al., 2016) also explored people’s experience of dog bites in the previous five years and the practice of bite treatment in these cases. This additional measure of actual dog bite frequency and treatment sought would have been beneficial in our study, as available bite data from local health providers lacks the geographic specificity to target only these three *desas*.

The KAP survey indicated dog owners had significantly more positive attitudes to dogs, responsible dog care and vaccination than non-dog owners. However, taken as one population, people’s attitudes were overall positive, with a preference for vaccination over culling to control rabies. However, there was a notable proportion of people, in particular non-dog owners, who agreed with the statement that “Mass culling of dogs is always necessary when there is one dog with rabies in my *banjar*”. This appears to be in contrary to the agreement with the other more positive statements about dogs and vaccination as opposed to culling. However, there had been significant publicity of the opinion that culling was a necessary part of the rabies response, despite evidence to the contrary [5,6,7]. Hence it appeared citizens had absorbed some of this messaging, in opposition to their generally more positive attitude to dogs; a discord that may have caused internal conflict.

Public messaging on rabies in Bali had also focused on how to treat dog bites and how to spot a suspect rabid dog. This messaging appears to have been well incorporated into people’s knowledge, as evidenced by the vast majority of accurate responses to the questions of what to do when bitten and what the rabies symptoms in dogs are. There was a minority of people that also mentioned using traditional medicine to treat a bite, however this may have not been their exclusive response, but alongside attendance at a bite clinic or hospital.

This study highlighted that effective rabies control was potentially very achievable for these *desas*. The low percentage of unowned dogs and positive responses of dogs to both owners and T1s suggested that vaccination campaigns could easily be achieved through central point and door-to-door methods with relatively limited need for expert catchers and handlers. Respondents to the KAP also displayed good knowledge of rabies and had a positive attitude to vaccination. The current vaccination coverage of 56% was already above the critical 20–45% threshold required to prevent the dog population acting as a reservoir for rabies [13], but below the 70% target for annual vaccination campaigns to maintain herd immunity above the critical threshold despite dog population turnover[3]. The strong presence of unvaccinated dogs (44%) shows the risk of short transmission chains of rabies following an incursion, something that could occur in this open border environment, especially considering that 37.5% of dogs were sourced from outside the *desa* (either purchased, received as a gift, or adopted from the street).

The challenges to rabies control include the high turnover of dogs. The age structure indicates young dogs represent the vast majority of the population and are most at risk of not being vaccinated. Abandonment of unwanted dogs is also a risk as these dogs are less likely to be vaccinated. Establishing an estimate of the number of dogs abandoned is difficult as people are unlikely to admit to this practice. The prior ownership status of the currently unowned dogs could not be definitively identified, however it was suspected that many were abandoned dogs. The welfare state of these unowned dogs was often poor; 23% were in a thin or emaciated body condition, 19% had visible skin conditions, and 10% had visible injuries. The poor visible welfare state of unowned dogs suggests their survival is poor hence even though owned dogs may be abandoned relatively frequently, they do not live long, so the population size of unowned dogs remains low.

## 5. Conclusions

The purpose of this study was to provide an evidence base for the design of a subsequent rabies and dog population management intervention. The dogs appear to be accessible for vaccination and people are primed to accept the need to vaccinate. Additional activities to reduce population turnover and increase the care provided to dogs should reduce the proportion of young susceptible dogs, roaming behaviour, and abandonment; this should increase vaccination coverage and the resilience of the dog population to any incursions of rabid dogs, hence protecting people from this fatal disease.

## Figures and Tables

**Figure 1 animals-08-00112-f001:**
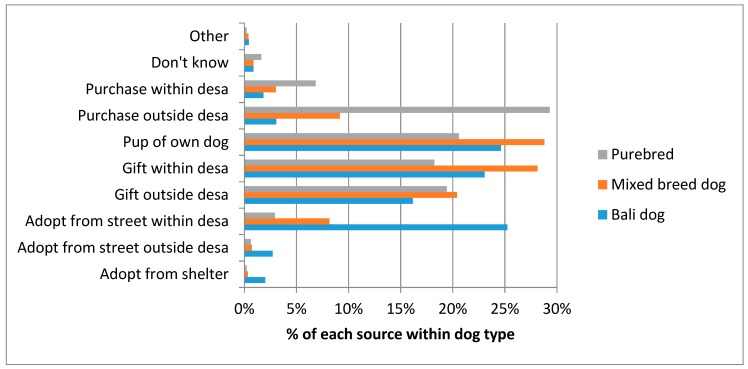
Source of dogs, broken down by dog breed type.

**Figure 2 animals-08-00112-f002:**
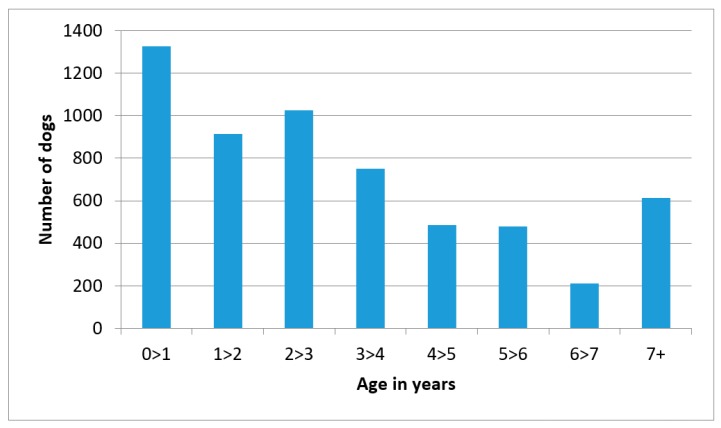
Ages of owned dogs.

**Figure 3 animals-08-00112-f003:**
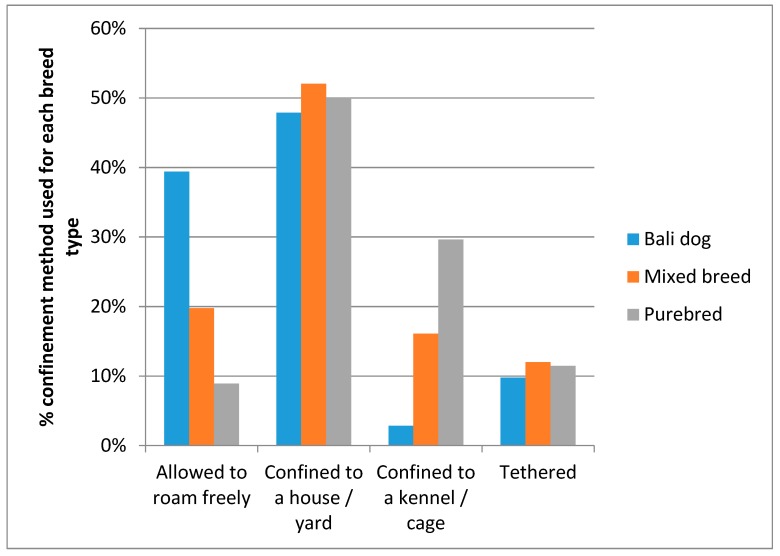
Confinement methods, separated by breed type.

**Figure 4 animals-08-00112-f004:**
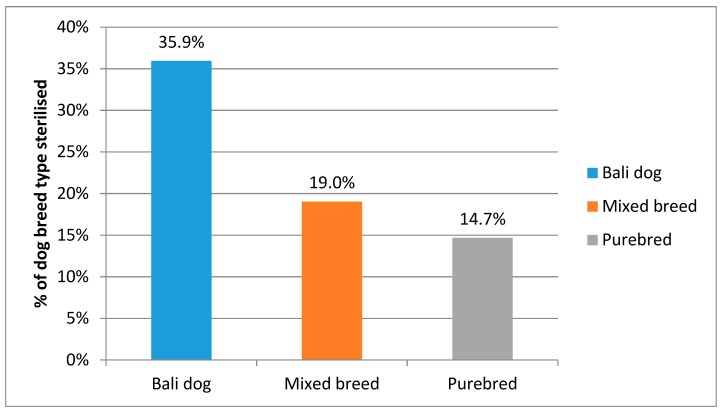
Percentage of each breed type that is sterilised.

**Figure 5 animals-08-00112-f005:**
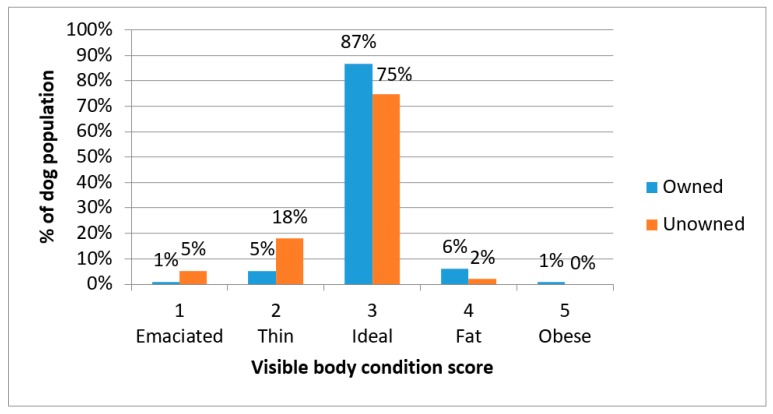
Percentage of owned and unowned dogs observed with body condition scores 1 (emaciated) through to 5 (obese).

**Figure 6 animals-08-00112-f006:**
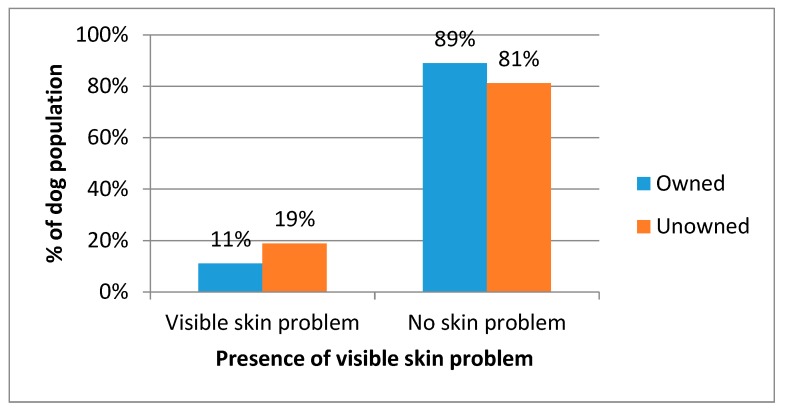
Percentage of owned and unowned dogs observed with and without a visible skin problem.

**Figure 7 animals-08-00112-f007:**
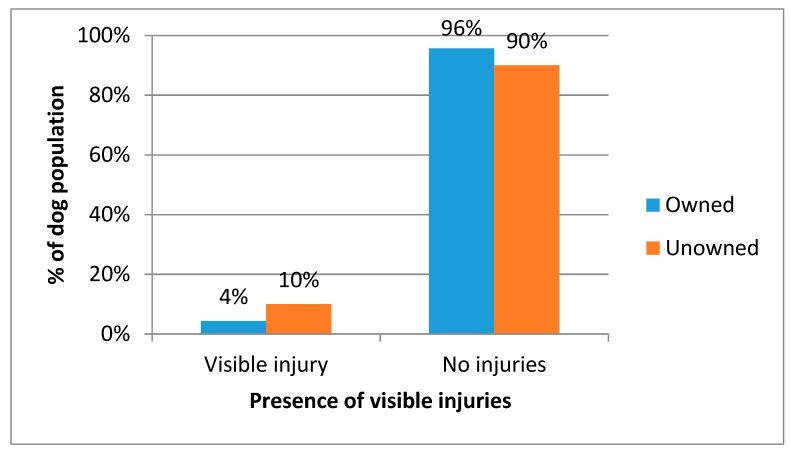
Percentage of owned and unowned dogs observed with and without a visible skin problem.

**Figure 8 animals-08-00112-f008:**
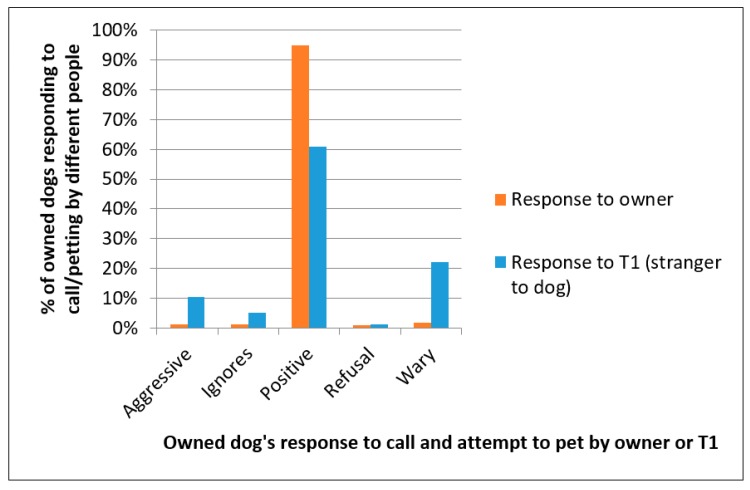
Responses of owned dogs to being called over and petted by owners and T1s.

**Figure 9 animals-08-00112-f009:**
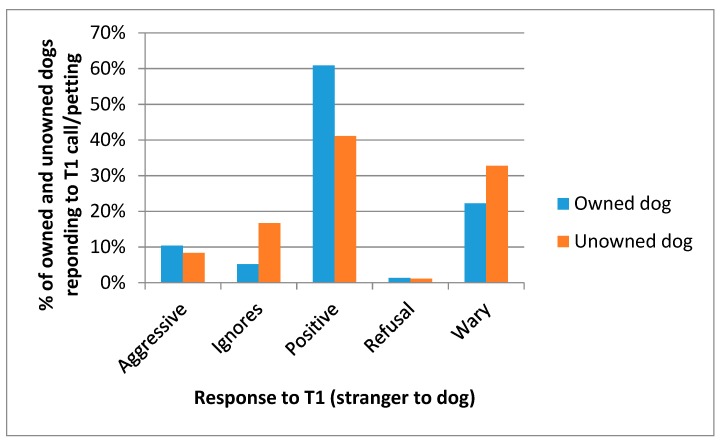
Responses of owned and unowned dogs to being called over and petted by T1s.

**Figure 10 animals-08-00112-f010:**
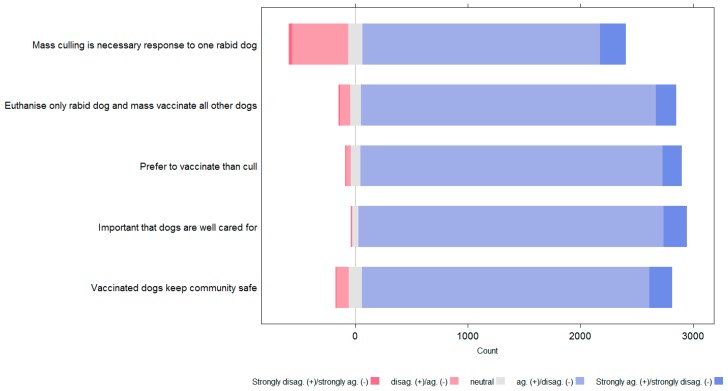
Level of dog owner agreement with attitude statements.

**Figure 11 animals-08-00112-f011:**
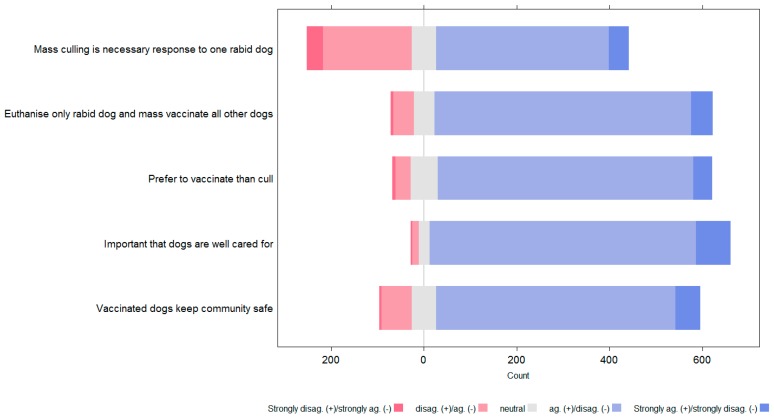
Level of non-owner agreement with attitude statements.

**Figure 12 animals-08-00112-f012:**
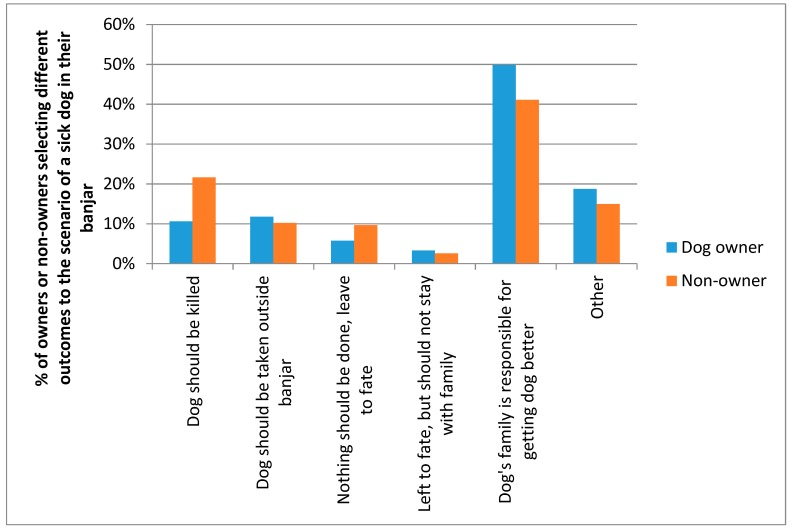
Dog owner and non-owner responses to what should be done in the scenario of a sick dog in their *banjar*.

**Figure 13 animals-08-00112-f013:**
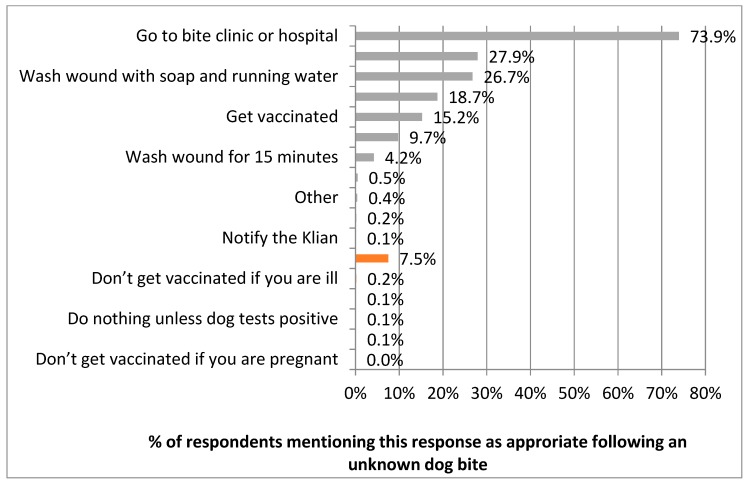
Respondents’ responses to the question of what are the appropriate actions following a bite from an unknown dog; correct actions are in green, potentially risky actions are in red.

**Figure 14 animals-08-00112-f014:**
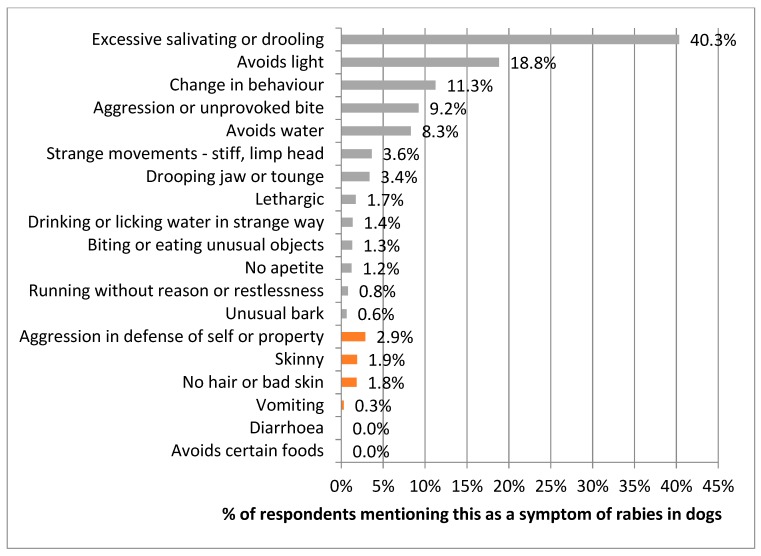
Respondents’ responses to the question of what symptoms are shown by rabid dogs; correct symptoms are in green, incorrect symptoms are in red.

**Figure 15 animals-08-00112-f015:**
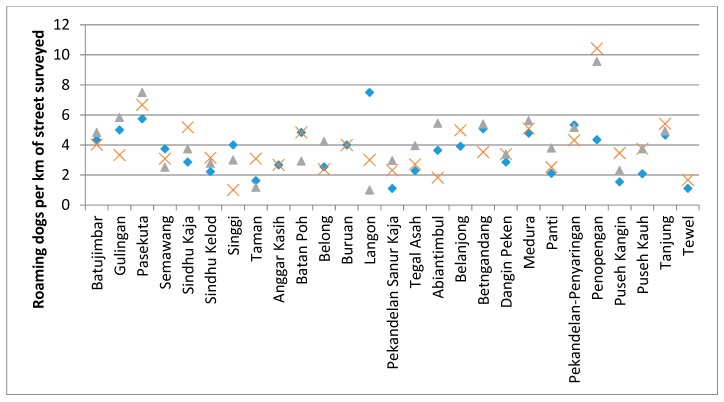
The number of roaming dogs per km of street surveyed is shown for each *banjar*. The three icons represent the density on each of the three replicate surveys along the same *banjar* route; blue diamond on day 1, green triangle on day 2, and red cross on day 3.

**Table 1 animals-08-00112-t001:** Odds ratio of not being vaccinated for each variable in a logistic regression main effects model with one level acting as the reference against which all other levels are compared.

Variable	Level	Vaccination Coverage	Odds Ratio	Std Error	*p*	Significance
Age	Reference ^1^: Old adult	67.5%				
	Puppy	16.0%	10.951	1.156	0.000	**^2^
	Juvenile	51.8%	1.908	1.130	0.000	**
	Adult	63.8%	1.182	1.104	0.091	
Breed type	Reference: Purebred dogs	62.5%				
	Bali dog	49.7%	1.348	1.086	0.000	**
	Mixed breed	52.1%	1.276	1.071	0.000	**
Confinement	Reference: Confined to house or yard	58.2%				
	Kennel	58.6%	1.082	1.084	0.327	
	Roaming	54.0%	1.286	1.084	0.002	*
	Tethered	59.0%	1.197	1.100	0.059	
Gender	Reference: Male	57.5%				
	Female	55.2%	1.082	1.061	0.181	

^1^ Reference denotes the level of the variable that the other levels are compared against when calculating the odds ratio. For example, the odds ratio for puppies is calculated from the odds of puppies not being vaccinated divided by the odds of old adults not being vaccinated. ^2^ Denotes level of significance: ** indicates a highly significant result where *p* < 0.001, * indicates a significant result where *p* < 0.05.

**Table 2 animals-08-00112-t002:** Full attitude statements and key phrases as used in Figure 10 and Figure 11.

Key Phrase	Full Attitude Statement
Mass culling is a necessary response to one rabid dog	“Mass culling of dogs is always necessary when there is one dog with rabies in my *banjar*”
Euthanise only rabid dog and mass vaccinate all other dogs	“Imagine a dog that looks like it has symptoms of rabies is found in your *banjar*. Only the dog with rabies should be culled, all other dogs should be vaccinated”
Prefer to vaccinate than cull	“I prefer to vaccinate all the dogs in their *banjar* rather than cull them”
Important that dogs are well cared for	“It is important to me that dogs in my *banjar* are well cared for”
Vaccinated dogs keep the community safe	“Vaccinated dogs keep the community safe”

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
