# Peer review of "Dog Ecology and Rabies Knowledge of Owners and Non-Owners in Sanur, A Sub-District of the Indonesian Island Province of Bali"

_animals, 2018, doi:10.3390/ani8070112_

Round 1

Reviewer 1 Report

Review Report                                           

ID Animals 314792

Title:

„Dog ecology and rabies knowledge of owners and non-owners in Sanur, a sub-district of the Indonesian island Province  of Bali

 Author‘s:  Elly Hiby, Kadek Karang Agustina, Kate Nattras Atema, Gusti Ngurah Bagus, Janice Girardi, Mike Harfoot, Yacinta Haryono, Lex Hiby, Hendra Irawan, Levin Kalalo, Sang Gede Purnama, I Made Subrata, Ida Bagus Ngurah Swacita, Ni Wayan Arya Utami, Pande Putu Januraga, Dewa Nyoman Wirawan

Version: 1 Date: 01/06/2018

Reviewer number: 1

A brief summary

The manuscript "Dog ecology and rabies knowledge of owners and  non-owners in Sanur, a sub-district of the Indonesian island Province  of Bali”

By Elly Hiby, Kadek Karang Agustina, Kate Nattras Atema, Gusti Ngurah Bagus, Janice Girardi, Mike Harfoot, Yacinta Haryono, Lex Hiby, Hendra Irawan, Levin Kalalo, Sang Gede Purnama, I Made Subrata, Ida Bagus Ngurah Swacita, Ni Wayan Arya Utami, Pande Putu Januraga, Dewa Nyoman Wirawan provides social investigation (the combination of census) and questionnaire exploring knowledge including practice relating to rabies in the light of dog demography status in three villages in Sanur, Bali.  The objective of the Article can be identified in the Introduction part - to investigate the structure of native dog population, local ownership practices and to improve responsible ownership knowledge as a one of rabies control priority. According to my opinion, the objectives of the article shall be more focus on the dog population management and the rabies prevention in restricted dog population including rabies vaccination and ownership practices. 

Broad comments

For a veterinary science point of view the manuscript is not original, but the basic questionnaire data of dog demography, street survey, and welfare status or vaccination coverage comparison can be significant, especially for the ecologist or epidemiologists in some regional countries where rabies is endemic. The Article is written in a relative appropriate way, but can be better structured – the paper contain a lot of different simple statistical data: gender ratio, dog breed types, ages of owned dogs, confinement, „sterilization“, welfare status, responsible ownership knowledge…All this information should be better explained – it is necessary to precisely identify and present how the all multiple information relates to the subject of the research; how a little linked metadata can be interpreted to meet the main (possible) objectives of the article - the rabies control measures in the relevant dog population. So, the first impression is that the topics, analyzed in the article, are unrelated…all the “cumulative” information and interpretation should be presented in the “Results” and the “Discussion” parts. If the questionnaire is used, the criteria for included different questions should be clearly identified (in “Materials and Methods” part).

The English language is rather appropriate and understandable.

The Introduction could more focus on the objective(s) of the study: more data about the rabies epidemiological situation and the prevention measure (in dog population) in the neighboring Indonesian regions should be included in the paper as well. At the end of “Introduction” the goal (not goals) should be clearly identified.

The Discussion part should be shortened (more than 50% if possible), in particular, on research on dog demographic questionnaires. The discussion (L354-416) about the dog demography questionnaire comparative analysis should be more directly associated with the rabies ctrl. The discussion from L417 - already better…All the abbreviations (OSM,  GLM, AIC,…) with explanations (used first time in the text) should be included in the manuscript.

The list of reference should be revised and unified according to the specific recommendation for authors. 

Specific comments

L35-36. „Simple Summary: That rabies can be managed humanely using vaccination rather than culling is well accepted...“. It is understandable what the authors wanted to say and in theory everything is in order, but the idea of the sentence does not sound ethically enough… I recommend that this sentence be corrected. I think “…That rabies can be managed humanely using vaccination is well accepted...“it would be better.

L58. Keywords. I recommend 4-6 single keywords (for example: canine, rabies; vaccination, owner knowledge, Bali).

L125-126. After “Epi Info or Device Magic in combination with Wise Monkey“ use the identification – Reference source, producers names, etc... Should be included.

L145. After „...OSM tracker“ use the identification – Reference source, producer’s names, etc... Should be included.

L149-150.  After „...analyzed using Epi Info Visual Dashboard„ use the identification – Reference source, producers names, etc... Should be included.

L155/L160. After „Chi-squared tests‘use the identification – Reference source should be included.

L156-157. After „...conducted in R using R package ‘multcomp’ „ use the identification – Reference source should be included.

L198. Figure 1. „This bar chart shows...“should be deleted.

L203.Figure 2. „Histogram showing...“should be deleted.

L213. Figure 3. „Bar chart showing ...“should be deleted.

L221. Figure 4. „Bar chart showing....“should be deleted.

L245-247. Table 1. Should be replaced by „Results of the logistic regression main effects model - the odds ratio variability of not vaccinated dogs“ or similar....

L266 /272/278/288/294/325. Figures 5/6/7/8/9/12. „Bar chart showing....“should be deleted.

L309/L312 Figures 9 and 10 „Diverging stacked bar charts displaying...“should be deleted (or all the figures titles should be overwrite).

L350. Figure 15. To identify what symbols means.

Author Response

Our thanks to Reviewer 1 for providing such thoughtful comments; we have implemented these and believe it has very much improved the manuscript. Specific changes are as follows:

- We have included a short addition on rabies epidemiology in Indonesia, but were not able to find much published literature on the national situation. However, in the methods, we have included more detail on rabies epidemiology in the area where the study took place.

- We feel that the goal of studying these dog and human populations to design an intervention is already clearly stated in the introduction.

- In the methods section, we have included a reference for where the questions were selected from.

- All abbreviations have been defined and referenced.

- The discussion has been shortened and emphasis has been put on the interpretation of this cumulative information for planning rabies control.

Response to specific comments:

·         L35-36. „Simple Summary: That rabies can be managed humanely using vaccination rather than culling is well accepted...“. It is understandable what the authors wanted to say and in theory everything is in order, but the idea of the sentence does not sound ethically enough… I recommend that this sentence be corrected. I think “…That rabies can be managed humanely using vaccination is well accepted...“it would be better.

-          Agreed, we have made this change.

·         L58. Keywords. I recommend 4-6 single keywords (for example: canine, rabies; vaccination, owner knowledge, Bali).

-          We agree that ‘owner knowledge’ is a good term and have adopted that. However, the limit on keywords is 10, we are under this limit at the moment and would prefer to maintain the additional words to help people with their literature searching in future.

·         L125-126. After “Epi Info or Device Magic in combination with Wise Monkey“ use the identification – Reference source, producers names, etc... Should be included.

-          Agreed, we have added this information, including website addresses.

·         L145. After „...OSM tracker“ use the identification – Reference source, producer’s names, etc... Should be included.

-          Agreed, we have added this information, including website addresses.

·         L149-150. After „...analyzed using Epi Info Visual Dashboard„ use the identification – Reference source, producers names, etc... Should be included.

-          Agreed, we have added this information, including website addresses.

·         L155/L160. After „Chi-squared tests‘use the identification – Reference source should be included.

-          We have not been able to find a suitable reference for this statistical test. As it is such a fundamental test we argue that most readers will be aware of its use.

·         L156-157. After „...conducted in R using R package ‘multcomp’ „ use the identification – Reference source should be included.

-          Agreed, we have added this information, including website addresses.

·         L198. Figure 1. „This bar chart shows...“should be deleted.; L203.Figure 2. „Histogram showing...“should be deleted.; L213. Figure 3. „Bar chart showing ...“should be deleted.; L221. Figure 4. „Bar chart showing....“should be deleted.

-          Agreed, this text has been deleted

·         L245-247. Table 1. Should be replaced by „Results of the logistic regression main effects model - the odds ratio variability of not vaccinated dogs“ or similar....

-          Agreed, reworded this title

·         L266 /272/278/288/294/325. Figures 5/6/7/8/9/12. „Bar chart showing....“should be deleted; L309/L312 Figures 9 and 10 „Diverging stacked bar charts displaying...“should be deleted (or all the figures titles should be overwrite).

-          Agreed, this text has been deleted.

·         L350. Figure 15. To identify what symbols means.

-          Agreed, an explanation has been added

Reviewer 2 Report

The authors should be congratulated for undertaking a tremendous effort to use robust methods to better understand the dog population in these communities. The statistical methods are rigorous and the results will surely help these communities to better understand how to manage their dog population and prevent rabies introductions. The manuscript is well written, however it could be improved through more thorough background for the readers on why these communities were chosen (particularly since they are not affected by canine rabies). Also, while the methods are results are impressive, I felt that the discussion section fell short of truly interpreting the results and describing the potential impact of these findings.  Please see comments below for suggestions to improve this aspect of the paper.

INTRODUCTION:

Line 66: for this point to be true the authors should clarify that “indiscriminate” culling has not worked in recent decades. Most rabies free countries practice large-scale, systematic, humane, "culling" of unwanted animals and OIE recommends population management as one of the core components of a comprehensive rabies control program.  As the authors point out in lines 84-85, in most cultures abandonment of unwanted dogs is an issue that hampers rabies control efforts; but what is not mentioned in this introduction is the use of humane, systematic removal of unwanted dogs from communities, which is common practice in many countries .  I recommend that the authors review the manuscript to ensure that this nuanced, yet critical, difference between indiscriminate and systematic culling is clear throughout the paper. The authors should also be sure to provide well-rounded evidence for rabies control in their discussion; as the practices of most rabies free countries includes shelter-based removal and euthanasia of unwanted animals is not even mentioned.

Is it possible to include more background on the current rabies situation in Bali, and the relation between the 2008 outbreak and these three desas that were surveyed?

METHODS:

Why were these desas chosen, particularly if they are not enzootic for canine rabies?

2.1 – More explanation of the census method is needed.

      - Please describe the timing of the household surveys in more detail. Were they conducted throughout the year, at certain timepoints throughout the year, a retrospective survey of dog ownership over the past year, or other methodology?

     - How many total household surveys were conducted?

     - How did you ensure that dogs claimed by one household were not also claimed by another household? 

     - How did you account for loosely owned or community dogs, that may receive partial care, but not be claimed by a single household?

Line 115: how did you confirm that a dog was “unowned”?

Line 115: how did you ensure that you did not collect information for an unowned dog more than once?

Line 143: could using motorbikes decrease identification of free roaming dogs?  Other papers describe walking through communities to improve the chances of detecting dogs. Could using motorbikes reduce your detection probability? and if so, how would that affect your estimates?

2.3 – what times of day were the sights conducted? What time of year were these surveys conducted?

Lin 161 – 162: analyzing dog population data by linear km or km2 requires the assumption that the areas are homogenous in terms of use, human density, ownership practices, etc. Please specify that analysis was conducted per linear km of road (rather than km2). Can the authors please provide more evidence in the methods or introduction to support that analysis by linear km is appropriate in these three settings?

RESULTS:

Please report on how many households were approached during the census, how many households consented to the survey, and how many allowed their data to be used. The methods state that data were removed if the household did not consent. To interpret a census we must know how many households were in the desas, how many were approached, and how many consented.

Line 182: please clarify in methods how dogs were determined to be owned

Line 184: please specify where the human population was obtained from. Is this from government records or was this collected as part of the census?

Line 201: do the authors truly believe there was a 20-year old dog in this cohort?

Line 205: please define a “yard” in the methods section. A dog in a yard that is not secure is not a confined dog.

Line 316: this interpretation is discussion, not a result.

Line 345: this should be in the methods

Line 345: if multiple days of street surveys were conducted, why was this data not used to calculated an estimated number of free roaming dogs, preferably via the Lincoln-Petersen equation?

DISCUSSION

Line 417 – 430: this entire paragraph reads more like a repeat of the results rather than discussion. What is the meaning of this findings, and what can this knowledge do to improve vaccination coverage?  Should vaccination campaigns be held more frequently?  Should vaccine be available in the community year-round, with a focus on vaccinating puppies?

Lines 431 – 439: what is the importance of knowing this information?  What is the interpretation?  Again, this paragraph reads more like another summary of the results rather than a discussion. What total proportion of dogs do the authors feel could be vaccinated without the use of capture equipment? What proportion would likely require advanced equipment, and how would that impact the current vaccination strategy?  Can this information be used to develop the "ideal" mixed-methods vaccination approach? 

Line 466 – please provide a reference for the 20-40% statement

Line 475 – Abandonment does not seem to be an issue in these communities, as only 3% of dogs were unowned. Please explain this in more detail.

Line 488 – what is the effective vaccination coverage needed to keep rabies out of a community?  That seems to be the premise of this study, since this is a community that has not been affected by rabies. However, the authors do not provide citations or evidence for what this level of “maintenance” coverage is.

Author Response

Our thanks to Reviewer 2 for providing such thoughtful comments; we have implemented most of these and believe they have very much improved the manuscript. Changes and comments are as follows:

·         Indiscriminate culling.

-          We disagree that the term ‘indiscriminate’ is required. The papers that we cite also do not talk only about indiscriminate culling, they argue that dog population reduction is not required for rabies control. Although we agree that humane and discriminate culling of unwanted dogs is conducted in many countries as part of dog population management with the goal of zero roaming dogs, this is not required for effective rabies control. There are examples of developing world locations that have achieved rabies control without culling or shelters, instead managing their street dog population in situ (e.g. Reece, J. F.; Chawla, S. K. Control of rabies in Jaipur, India, by the sterilisation and vaccination of neighbourhood dogs. Vet. Rec. 2006, 159, 379–383). The project that has been conducted in Sanur, Bali since this baseline study was completed has not used culling at all; it has euthanased sick dogs, but left in situ healthy unowned dogs after sterilisation and vaccination, in collaboration with the local community. Shelter-based removal and euthanasia of unwanted animals appears neither economically viable nor desired by Sanur communities at this time; their goal is rabies control as opposed to dog population management to the point of no roaming dogs.

·         Why were these desas chosen?

-          We have added explanation re the choice of these desas at the beginning of the methods section.

·         Please describe the timing of the household surveys in more detail. Were they conducted throughout the year, at certain timepoints throughout the year, a retrospective survey of dog ownership over the past year, or other methodology?

-          We have added text to the methods section to answer these questions

·         How many total household surveys were conducted?

-          This is not known. The dogs were the sampling unit rather than the household. Although we recorded the name of owner and therefore know how many owners we interviewed, there were sometimes more than one owner living in a household. Some Balinese households are comprised of extended families living in different buildings within one compound, making the definition of ‘household’ complex.

·         How did you ensure that dogs claimed by one household were not also claimed by another household?

-          We have explained more about the use of our photo catalogue in the text; we felt that this worked well because Bali dogs are thankfully quite varied in appearance.

·         How did you account for loosely owned or community dogs, that may receive partial care, but not be claimed by a single household?

-          Loosely owned dogs were defined as owned and their owners (may have been more than one if they lived between households, as could happen in an extended family compound of several dwellings) interviewed in the same way as a well confined ‘pet’ dog; their answers about levels of care, in particular confinement would obviously be different. Community dogs were defined as unowned and community members, often more than 1, interviewed about the dogs to establish their age and if it had been sterilised or vaccinated in the past.

·         Line 115: how did you confirm that a dog was “unowned”?

-          As described in the text, any apparently unowned dog became the subject of conversation with people living and working locally, this is a close knit community and people know a lot about each other’s animals. We also used photographs to check if these dogs had been seen before – we have added text to explain this bit.

·         Line 115: how did you ensure that you did not collect information for an unowned dog more than once?

-          By using photographs and by allocating banjars to 1 or 2 T1s, as now explained in the text in section 2.1.

·         Line 143: could using motorbikes decrease identification of free roaming dogs? Other papers describe walking through communities to improve the chances of detecting dogs. Could using motorbikes reduce your detection probability? and if so, how would that affect your estimates?

-          Motorbikes are an excellent mode of transport for this kind of survey. They are easy to stop and manoeuvre if you need to get a better look at a dog (unlike driving a car) but they also ensure that you are moving faster than the dogs themselves, so you reduce double counting – a significant risk with walking surveys. These surveys were to measure a change in density of roaming dogs on the street and their visible welfare state over time, not to produce an estimate of the number of dogs. The most important aspect of such surveys is that they are done using the same protocol and therefore detection probability over time, so you see a consistent proportion of the roaming dogs on the street. They also needed to be efficient as we wanted these surveys to be repeated at least twice a year for the length of the project – walking would have been too time consuming for these teams given the lengths of street involved. This has been explained more fully in the text in section 2.3.

·         What times of day were the sights conducted? What time of year were these surveys conducted?

-          This has been added to section 2.3.

·         Lin 161 – 162: analyzing dog population data by linear km or km2 requires the assumption that the areas are homogenous in terms of use, human density, ownership practices, etc. Please specify that analysis was conducted per linear km of road (rather than km2). Can the authors please provide more evidence in the methods or introduction to support that analysis by linear km is appropriate in these three settings?

-          We make no assumptions about the homogenous nature of street use, we know this is not the case. This survey is not designed to estimate the roaming dog population size, rather a measure of density expressed as the number of roaming dogs observed per km of street surveyed. Our ambition, and what has come to pass, was to monitor changes in dog density over time. We have explained this more in the text in section 2.3, including a reference to a paper that positions this is a valid method of monitoring dog populations.

·         Please report on how many households were approached during the census, how many households consented to the survey, and how many allowed their data to be used. The methods state that data were removed if the household did not consent. To interpret a census we must know how many households were in the desas, how many were approached, and how many consented.

-          We are unable to present this data. Unfortunately the number of households that did not consent to take part was not recorded consistently.

-          We have added the number of owners interviewed during the census and therefore the number of dogs per owner. Due to the structure of households in Sanur, this may not equate exactly to the number of dogs per households. We have explained this further in section 3.1.

·         Line 182: please clarify in methods how dogs were determined to be owned

-          Amended in manuscript.

·         Line 184: please specify where the human population was obtained from. Is this from government records or was this collected as part of the census?

-          Amended in manuscript.

·         Line 201: do the authors truly believe there was a 20-year old dog in this cohort?

-          We agree it is unusual! But apparently the owner was very sure of the age.

·         Line 205: please define a “yard” in the methods section. A dog in a yard that is not secure is not a confined dog.

-          This has been defined and clarified in the section 2.1

·         Line 316: this interpretation is discussion, not a result.

-          We agree. We have removed this from the results section; it was already included in the discussion section.

·         Line 345: this should be in the methods

-          We agree. It has been relocated to section 2.3.

·         Line 345: if multiple days of street surveys were conducted, why was this data not used to calculated an estimated number of free roaming dogs, preferably via the Lincoln-Petersen equation?    

-          The purpose of this survey was to establish baseline density of roaming dogs on the street and not an estimate of the roaming dog population size. Methods of accurately estimating population size are more time consuming and would not have been viable for this team to repeat over time (see the Hiby and Hiby 2017 referenced in the manuscript for further discussion of this). There was no marking used during the questionnaire, nor was marking envisaged for the subsequent intervention, so using a mark-recapture method such as a Lincoln-Petersen was not viable, as it would have required a specific marking effort for the street survey which was beyond the available time resources.

·         Line 417 – 430: this entire paragraph reads more like a repeat of the results rather than discussion. What is the meaning of this findings, and what can this knowledge do to improve vaccination coverage? Should vaccination campaigns be held more frequently? Should vaccine be available in the community year-round, with a focus on vaccinating puppies?

-          Agreed. We have included additional explanation of how these results were used in the intervention design.

·         Lines 431 – 439: what is the importance of knowing this information? What is the interpretation? Again, this paragraph reads more like another summary of the results rather than a discussion. What total proportion of dogs do the authors feel could be vaccinated without the use of capture equipment? What proportion would likely require advanced equipment, and how would that impact the current vaccination strategy? Can this information be used to develop the "ideal" mixed-methods vaccination approach?

-          Agreed. This para has been amended.

·         Line 466 – please provide a reference for the 20-40% statement

-          Amended in manuscript.

·         Line 475 – Abandonment does not seem to be an issue in these communities, as only 3% of dogs were unowned. Please explain this in more detail.

-          Amended in manuscript.

·         Line 488 – what is the effective vaccination coverage needed to keep rabies out of a community? That seems to be the premise of this study, since this is a community that has not been affected by rabies. However, the authors do not provide citations or evidence for what this level of “maintenance” coverage is.

-          We have used the WHO recommended 70% vaccination for annual campaigns as the target coverage – we have now added a reference for this 70%. We are planning a more detailed discussion on coverage for a subsequent paper presenting the results of the intervention.